# Primitive Chain Network Simulations for Double Peaks in Shear Stress under Fast Flows of Bidisperse Entangled Polymers

**DOI:** 10.3390/polym16111455

**Published:** 2024-05-21

**Authors:** Yuichi Masubuchi

**Affiliations:** Department of Materials Physics, Nagoya University, Nagoya 464-8603, Japan; mas@mp.pse.nagoya-u.ac.jp

**Keywords:** polymer dynamics, viscoelasticity, coarse-grained molecular simulations, entangled polymers, rheology

## Abstract

A few experiments have reported that the time development of shear stress under fast-startup shear deformations exhibits double peaks before reaching a steady state for bimodal blends of entangled linear polymers under specific conditions. To understand this phenomenon, multi-chain slip-link simulations, based on the primitive chain network model, were conducted on the literature data of a bimodal polystyrene solution. Owing to reasonable agreement between their data and our simulation results, the stress was decomposed into contributions from long- and short-chain components and decoupled into segment number, stretch, and orientation. The analysis revealed that the first and second peaks correspond to the short-chain orientation and the long-chain stretch, respectively. The results also implied that the peak positions are not affected by the mixing of short and long chains, although the intensity of the second peak depends on mixing conditions in a complicated manner.

## 1. Introduction

Viscosity increase under high shear of entangled polymers has been widely investigated as a typical example of non-linear response [1,2,3,4,5,6]. Due to elasticity, viscosity initially increases linearly against applied strain, irrespective of the strain rate. When the shear rate is smaller than the slowest relaxation rate (the reciprocal longest relaxation time), viscosity gradually reaches the zero-shear value. Under high shear, steady-state viscosity decreases with increasing shear rate, exhibiting shear thinning. Before reaching a steady state, viscosity exhibits an overshoot. When the shear rate is lower than the reciprocal Rouse time, the peak is located at a total shear strain of ca. 2.3, reflecting segment orientation [7]. The total strain at the peak increases due to the contribution of chain stretch when increasing the shear rate. In some of the literature, viscosity exhibits an undershoot following the overshoot, which is related to the coherent tumbling of molecules [8,9,10,11]. Under extremely high shear, strain hardening without a steady state has also been reported [12,13]. 

Concerning viscosity overshoot, some of the literature reports multiple peaks rather than the widely observed single peak. Kinouchi et al. [14] reported viscosity growth curves with two peaks for polystyrene solutions with bidisperse molecular weight distributions in which a small amount of the long-chain portion was dispersed in a short-chain matrix. Osaki et al. [15] performed an extended study that varied the molecular weight of the short-chain. They reported that double-peak behavior is observed only when molecular weights are sufficiently separated. Snijkers et al. [16] reported multiple peaks for the viscosity growth of a comb polymer melt. These multiple overshoots may be attributable to flow instabilities and edge fracture [17,18,19]. However, Snijkers et al. [16] performed their experiments with a cone-partitioned plate rheometry [19] to exclude possible edge effects. 

The molecular origin of multiple peaks has yet to be discussed. Concerning bidisperse blends of linear polymers, Kinouchi et al. [14] and Osaki et al. [15] speculated that each mixed portion exhibits its own peak. Isram [20] theoretically supported this idea using his tube model. He concluded that the first and second peaks reflect overshoots in the stretch of short- and long-chain fractions, respectively. However, his theoretical model needs to be further considered, particularly with respect to the effects of constraint release. In the model, coupling between different fractions is accounted for by the chain contraction through convective constraint release; relaxation time is modified only under fast flow via chain stretch. In this implementation, the contribution of segment orientation may need to be considered further. Nevertheless, the comparison of the data by Osaki et al. [15] was not attained straightforwardly, and no discussion of segment orientation was given. Other molecular theories that predict the non-linear rheology of mixtures [21,22,23] have not been applied to this problem.

This paper examines the double peaks in viscosity growth under high shear for the bidisperse polystyrene solution reported by Osaki et al. [15], employing a multi-chain slip-link simulation [24] in which coupling between different components is directly considered. The results demonstrated that the second peak was due to the long-chain stretch, as Isram showed [20]. However, the first peak was attributable to segment orientation, and the contribution from the short-chain stretch was minor. Details are explained below. 

## 2. Model and Simulations

Because the model employed in this study was the same as that used in previous studies for shear flows [25,26,27,28,29,30], only a brief description is given below. In the primitive chain network model, a network consisting of nodes, strands, and dangling ends represents an entangled polymeric liquid. Each polymer chain in the system corresponds to a path connecting dangling ends through nodes and strands. This path is conceptually equivalent to the primitive path in the tube model [31]. However, no tube is considered in this model, and the path fluctuates due to force balance at each network node and dangling end. Namely, the positions of nodes and dangling ends obey a Langevin-type equation of motion, in which the force balance among the drag force, the tension acting on each strand, the osmotic force suppressing density fluctuations, and the thermal random force is considered. Mimicking entanglements, a slip-link is located at each node to bundle two subchains. The chains slide through slip-links, and chain sliding is described by the change-rate equation of the number of Kuhn segments on each strand according to the force balance for the node position along the chain backbone. Because of chain sliding, when a dangling end protrudes from the connected slip-link beyond a critical Kuhn segment number, a new node with a slip-link is created by hooking another segment randomly chosen from its surroundings. Alternatively, when a dangling end slides off the connected slip-link, the slip-link disappears, releasing the bundled chains. Concerning simulations under shear flows, infinitesimal-step shear strain is applied affinely to the network nodes and dangling ends, followed by relaxations according to the force balance mentioned above. This process of deformation and relaxation is repeatedly applied to attain a designated shear rate. Namely, for a required shear rate $\dot{\gamma }$, the applied step strain ∆γ is given as $∆\gamma =\dot{\gamma }∆t$, where ∆t is the numerical integration of step size. The length, energy, and time units chosen were the average strand length under equilibrium *a*, thermal energy kT, and the diffusion time of the node $\tau_{0}=\zeta a^{2}/6kT$, where ζ is the friction coefficient of the node. For convenience in converting to the experimental system, instead of a and kT, units of molecular weight $M_{0}$ and modulus $G_{0}$ were used. Here, $M_{0}$ is the average molecular weight of the strands, and $G_{0}=kT/a^{3}$. As discussed earlier, $M_{0}$ and $G_{0}$ are similar but different from the entanglement molecular weight $M_{e}$ and plateau modulus $G_{N}$. Nevertheless, the parameters $M_{0}$, $G_{0}$, and $\tau_{0}$ were determined according to the following procedure. $M_{0}$ was estimated from the entanglement molecular weight $M_{e}$ as $M_{0}=2M_{e}/3$, according to an empirical relation established earlier [32,33,34]. Using $M_{0}$, the average strand number per chain under equilibrium $Z_{0}$ was determined by $Z_{0}=M/M_{0}$, where M is the molecular weight of the examined polymer. A primitive chain network was created according to this $Z_{0}$, and an equilibrium simulation was conducted for a sufficiently long period, at least ten times longer than the longest relaxation time of the system. From stress fluctuations under equilibrium, the linear relaxation modulus G(t) was calculated with the Green–Kubo formula. The obtained G(t) was converted into G′(ω) and G″(ω) using REPTATE software (ver. 1.0.0 for Mac) [35,36]. These calculations were performed with dimensionless units, and the obtained moduli were converted into the experimental value via $G_{0}$, which was determined from $M_{0}$ by $M_{0}=\rho RT/G_{0}$. Here, ρ is the polymer density. Finally, $\tau_{0}$ was determined by fitting G′(ω) and G″(ω) to the experimental data, as demonstrated later. 

The system examined was a polystyrene tricresyl phosphate solution with a polymer concentration of 0.11 g/cm^3^. $M_{0}$, $G_{0}$, and $\tau_{0}$ were set at 10^5^, 2.5 kPa, and 0.12 s, respectively. The molecular weights of the short and long chains were $M_{L}=7.1\times 10^{5}$ and $M_{H}=8.2\times 10^{6}$, and these were replaced by chains with the segment numbers per chain of $Z_{L}=7$ and $Z_{H}=82$. Hereafter, subscripts L and H denote the low- and high-molecular-weight components; thus, L does not refer to the long-chain component but the short one. According to the experiment in [15], the volume fractions for the long and short chains were 0.09 and 0.91. This specific combination is referred to as f80/f850 in the work by Osaki et al. [15]. They mixed the polymer and solvent by adding dichloromethane, which was removed at 50 °C under lowered pressure. Rheological measurements were performed with a standard rotational rheometer (ARES). Dynamic viscoelastic tests were conducted with parallel plate fixtures with radii of 25 mm and 8 mm in a temperature range between −40 °C and 30 °C, and the viscoelastic master curve was constructed from the accumulated data via the time-temperature superposition for the reference temperature of 0 °C. The start-up shear response was measured under a cone-plate fixture with a radius of 25 mm and a cone angle of 6°. Measurements were performed at several temperatures chosen between −10 °C and 20 °C, and the obtained stress and applied shear rates were converted into the values at 0 °C according to shift factors determined via the linear response. 

Simulations were performed with a cubic simulation box with periodic boundary conditions. The Lees–Edwards boundary was employed to determine the shear gradient direction for the cases under shear flows. The box size was (16*a*)^3^, which is sufficiently larger than the radius of gyration of the examined chains under equilibrium. The segment density chosen was 10. See Figure 1 for a typical snapshot of the single long chain. Because of the large $M_{0}$, no impact from finite chain extensibility [37] was observed and not considered below. Friction reduction [38,39] was also excluded because the polymer concentration was low. Eight independent simulation runs were conducted for each shear rate, starting with different initial configurations; quantities reported below are ensemble averages taken during these simulations. For comparison, simulations for pure long- and short-chain systems were also performed with the same conditions, including segment density. 

Concerning the segment density in the simulations, since the value affects the entanglement network structure, it should be determined via the fitting of structural distribution functions such as the radial distributions of entanglement nodes [40,41]. However, this distribution was not available for the examined system, and thus, the standard value of 10, which has been utilized for various melts and solutions [8,9,25,26,27,28,29,30], was employed instead. Nevertheless, segment density did not affect viscoelasticity significantly [40]. 

## 3. Results and Discussion

Figure 2 shows the linear viscoelastic response, demonstrating the determination of $G_{0}$ and $\tau_{0}$. The result of the simulation performed with non-dimensional values are plotted against the top and right axes, while the experimental data are shown on the bottom and left axes. Parameters can be determined by shifting the simulation results vertically and horizontally. The simulation reasonably captures two plateaus in G′(ω) that reflect entanglements between all the chains and only between the long chains at the chosen parameter of $G_{0}=2.5\;kPa$. In contrast, the value $\tau_{0}=0.12\;s$ attains agreement only in the high-frequency regime; both red curves (from the simulation) overlap with symbols (from the experiment) in the range $\omega a_{T}>0.1$. However, the curves deviate from the data in the low-frequency range, implying that the longest relaxation time was overestimated. The fitting here was conducted to reproduce short-time (high-frequency) behavior, as in previous studies. One may vary $\tau_{0}$ to attain better agreement in the low-frequency range, sacrificing coincidence in the high-frequency range. Note that G″(ω) in the high-frequency range was also discrepant from the data, as it was underestimated due to the lack of higher Rouse modes in the simulation.

The specific reason for the discrepancy seen in Figure 2 is unknown. A possible reason is the effect of solvent quality with the use of time–temperature superposition to construct the experimental master curve. Since solubility depends on temperature, the vertical and horizontal shift factors used in the superposition might be complicated. Further discussion of this issue is difficult since the shift factors are not shown in the literature. Meanwhile, the simulation may have flaws, but previous studies have attained semi-quantitative agreement for bidisperse polystyrene melts [42,43]. 

In Figure 3a,b show the time development of shear stress σ and the first normal stress difference $N_{1}$ for some shear rates. For low shear rates, both σ and $N_{1}$ were quantitatively reproduced by the simulation despite the discrepancy in Figure 2. As the shear rate increased, the discrepancy between the data and the simulation results worsened. Concerning the shear stress shown in Figure 3a, the time development up to 10 s is well captured even under high shear. However, after exhibiting the first peak, the second peak in the simulation was weaker and occurred earlier than that observed experimentally. Since double-peak behavior was not apparent in the simulation, even for the highest shear rate reported experimentally, the result for a higher shear was added, as shown by the blue curve, which exhibits double peaks.

After bumping, the shear stress reached a steady value that was underestimated in the simulation. This discrepancy in the steady value is summarized in Figure 3c, which shows the steady-state viscosity plotted against the shear rate. The simulation results (red triangle) were smaller than those of the data (black circle) for the entire range of examined shear rates. However, the simulation was consistent with the Cox–Merz rule; the steady-state viscosity agreed with the complex viscosity (red broken curve). The viscosity obtained experimentally is located beyond the complex viscosity (black broken curve). Nevertheless, the discrepancy in the steady-state viscosity is not significant in the logarithmic plot in Figure 3a,c. The complex viscosity shown by the broken curves exhibits discrepancies consistent with Figure 1; the red curve obtained from the simulations overestimates the experimental value indicated by the black curve in the low-frequency range. In contrast, the curves overlap in the high-frequency range, where the start-up behavior shows reasonable agreement.

The first normal stress difference shown in Figure 3b demonstrates that these data are not accurately reproduced when the shear rate is high. In a short period of up to 20 s, the simulation reproduces the data irrespective of the shear rate. However, the simulation reaches its maximum at an $N_{1}$ value significantly lower than that in the experiment. The steady-state value is also underestimated. Note that the blue curve is the simulation result for $\dot{\gamma }$ = 2.5 s^−1^, for which the experiment was not performed, and the coincidence with the data at $\dot{\gamma }$ = 1.165 s^−1^ (uppermost symbols) is meaningless. Nevertheless, $N_{1}$ exhibits a single peak, even under high shear, where the shear stress shows double peaks. See the blue curves in Figure 3a,b. 

The discrepancy between the data and the simulation seen in Figure 3 is probably due to flaws in the simulation. For example, the simulation neglects hydrodynamic interaction, which comes into play for high shear when the shear rate is larger than the relaxation rate of the hydrodynamic blob. Tension acts on each segment is based on Gaussian chain statistics, irrespective of flows, but it should be modified according to the disturbed subchain distribution function. The other possibility is the creation and destruction of entanglement at the chain end. Based on the local equilibrium assumption, the rules are unchanged by flows, but segment orientation and stretch may affect them, particularly in entanglements between long and short chains. Nevertheless, the specific reason is unknown. 

Although the simulation does not reproduce previous data excellently, the origin of the double peaks in the shear stress is analyzed below for the case of the blue curve at the shear rate of 2.5 s^−1^ shown in Figure 3a. Figure 4 shows the contributions from the two components of shear stress, as well as the first normal stress difference compared to the response from the entire system (blue) and the results of the simulations without mixing. Concerning the double peaks for shear stress seen in panel (a), the first and second peaks come from short (green) and long (red) components, respectively, as speculated by Kinouchi et al. [14] The short-chain contribution shown by the solid green curve coincides with that obtained for the pure short-chain system displayed by the green broken-line curve, implying that mixing with the long chain does not affect short-chain behavior. In contrast, the mixing suppresses the peak stress for the long-chain contribution, although the peak position is unchanged. See the red curves. For the first normal stress difference, the peak originates from the long component, while the contribution from the short chains appears as a shoulder in the short-time region. The effects of mixing in short- and long-chain contributions are similar to those observed for shear stress. 

Further analysis of the molecular mechanism is shown below, according to a decoupling approximation [7,26], in which the stress was approximated as follows. Note that the non-linear spring constant is not considered because finite chain extensibility is excluded from this study.
$$\sigma \approx 3G_{0}\left(\frac{Z}{Z_{0}}\right)\lambda^{2}S$$
where σ is the stress tensor, Z is the number of segments under deformations, $\lambda^{2}$ is the squared segment stretch, and S is the segment orientation tensor calculated from the segment orientation vector u as $S_{\alpha \beta }=\langle u_{\alpha }u_{\beta }\rangle$. Figure 5 shows the molecular quantities for each component compared to the cases of the long and short chains without mixing. 

Concerning the short chain, all the examined quantities are insensitive to mixing with the long chain, as seen in the coincidence between solid and broken-line green curves. The peak in shear stress corresponds to the shear component of the orientation tensor $S_{xy}$ shown in panel (c), and it is not related to the $\lambda^{2}$ stretch in panel (b), according to the peak position. The peak of $S_{xy}$ is located at the applied shear strain of ca. 2.3, which is consistent with the tube theory, and no peak is observed for $\lambda^{2}$. The change in Z is relatively minor, slightly decreasing with time. The first normal stress, shown in Figure 4b, is consistent with the orientation of $S_{xx}{-S}_{yy}$ and $\lambda^{2}$; both monotonically increase and reach a plateau around $t/\tau_{0}\sim 20$. 

For the long chain, mixing with the short chain suppresses the magnitudes of Z reduction and $\lambda^{2}$ enhancement, as seen in Figure 4a,b. However, the characteristic times of their development are unchanged. For instance, the peak position of $\lambda^{2}$ and the mitigation times of $\lambda^{2}$ and Z are insensitive to mixing. These results are rationalized by the fact that the Rouse relaxation and chain contraction are insensitive to the change in the entanglement network. For the $S_{xy}$ shown in Figure 4c, the peak position is not affected by mixing and is located at the same strain as the short chain. An interesting feature is the undershoot following the peak enhanced by mixing, which reflects the coherent tumbling of the long chains [8,9]. However, the undershoot does not appear in shear stress because the increase in the segment stretch conceals it. Indeed, the position of the undershoot in $S_{xy}$ is close to that in $\lambda^{2}$. For the orientation $S_{xx}{-S}_{yy}$, no effect of mixing is seen, and thus the reduction in $N_{1}$ due to mixing in Figure 4b originates from the reduction in $\lambda^{2}$ in Figure 4b. 

Based upon the analysis above, the mechanism of double peaks in the shear stress of the bidisperse blend can be summarized as follows. Since each peak originates from the response of each component, as shown in Figure 4a, the first necessary condition is a separation of the peak positions of the two components. This condition is fulfilled if the Rouse times of two components are well separated and the applied shear rate is larger than at least one of the reciprocal Rouse times, since the peak is delayed due to the emergence of chain stretch. In the examined case, these are $\tau_{R}/\tau_{0}={Z_{0}}^{2}/2\pi^{2}\sim$2.5 and 341 for the short and long chains, respectively [44]. The normalized reciprocal Rouse times $\tau_{0}/\tau_{R}$ are then 0.4 and 2.9 $\times \;10^{-3}$. These values are not affected by mixing because chain stretch occurs irrespective of entanglement. The applied shear rate is $\dot{\gamma }\tau_{0}=0.3$, which is larger than $\tau_{0}/\tau_{R}$ for the long chain but smaller than that for the short chain. Thus, the short-chain stretch does not affect the first peak, which is dominated by the orientation of segments. In contrast, the second peak, caused by the long chain, is well delayed due to the long-chain stretch. As a result, two well-separated peaks are observed. In addition to the first condition concerning peak positions, the second necessary condition is that the magnitude of peaks must be comparable; if the second peak from the long chain component is too large, the first peak is concealed, and vice versa. Long-chain stress is reduced by mixing, due to the reduced long-chain fraction. However, mixing also induces the suppression of the long-chain stretch by culling the entanglements between long chains, and the corresponding stress overshoot is diminished. See Figure 4a. For the examined case, due to this suppressed long-chain stretch, the second peak is somewhat smeared. Consequently, the balance of molecular weights and volume fractions between two components is difficult to achieve. The explanation above is consistent with earlier conjectures [14,15] and theoretical analysis [20]. Nevertheless, the behaviors of molecular parameters in addition to chain stretch have now been shown for the first time. 

## 4. Conclusions

The molecular origin of double peaks in the time development of shear stress under fast shear of a bidisperse polystyrene solution, as reported by Osaki et al. [15], was analyzed using primitive chain network simulations. Their experimental data for linear and non-linear viscoelastic responses under shear were qualitatively reproduced, including the double peaks. Owing to the nature of multi-chain simulation, the entire stress was then decomposed into contributions from different components, and the results demonstrated that the first and second peaks came from the short- and long-chain components, respectively. Decoupling analysis of stress for each component revealed that the first and second peaks originated from the short-chain orientation and the long-chain stretch, respectively. Although these results are qualitatively consistent with earlier studies, they also imply that the necessary conditions for the emergence of double peaks are not easily fulfilled. Specifically, the condition is not trivial in realizing comparable peak intensities by balancing molecular weights and volume fractions between long- and short-chain components. This complication is probably why earlier reports about double peaks are few, apart from those discussing experimental difficulties. Further experiments and simulations to clarify such conditions are worth conducting. A similar analysis to understand responses under large-amplitude oscillatory shear [45] would be interesting. The results from supplemental studies will be published elsewhere. 

## Figures and Tables

**Figure 1 polymers-16-01455-f001:**
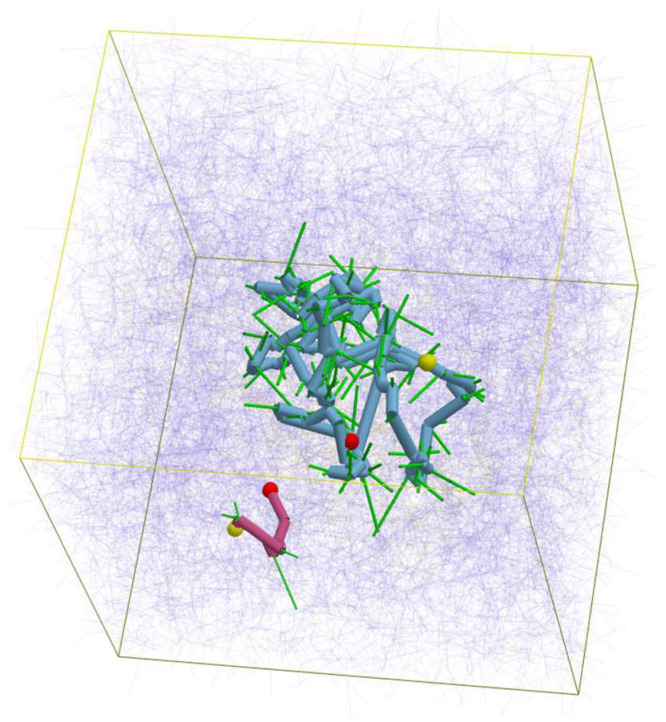
A typical snapshot of the system. Consecutive blue and red cylinders show a pair of long and short chains. Red and yellow spheres indicate the chain ends. Bold green lines show the entangled segments of the indicated chains. Thin blue lines show other segments. The simulation box is displayed as the yellow frame cube.

**Figure 2 polymers-16-01455-f002:**
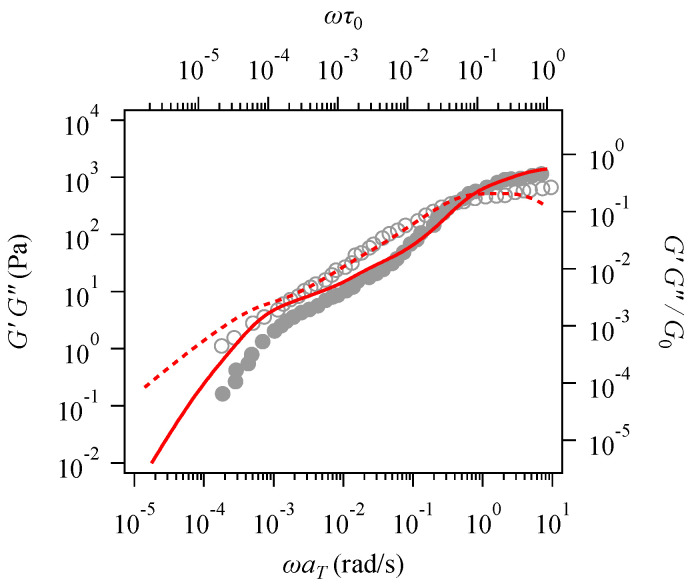
The linear viscoelasticity of the examined bidisperse polystyrene solution at 0 °C. The symbols represent experimental data from the literature [15] plotted against the left and bottom axes. Red curves show the simulation results from this study. The filled circle and solid curve are G′(ω), and the unfilled circle and broken-line curve are G″(ω).

**Figure 3 polymers-16-01455-f003:**
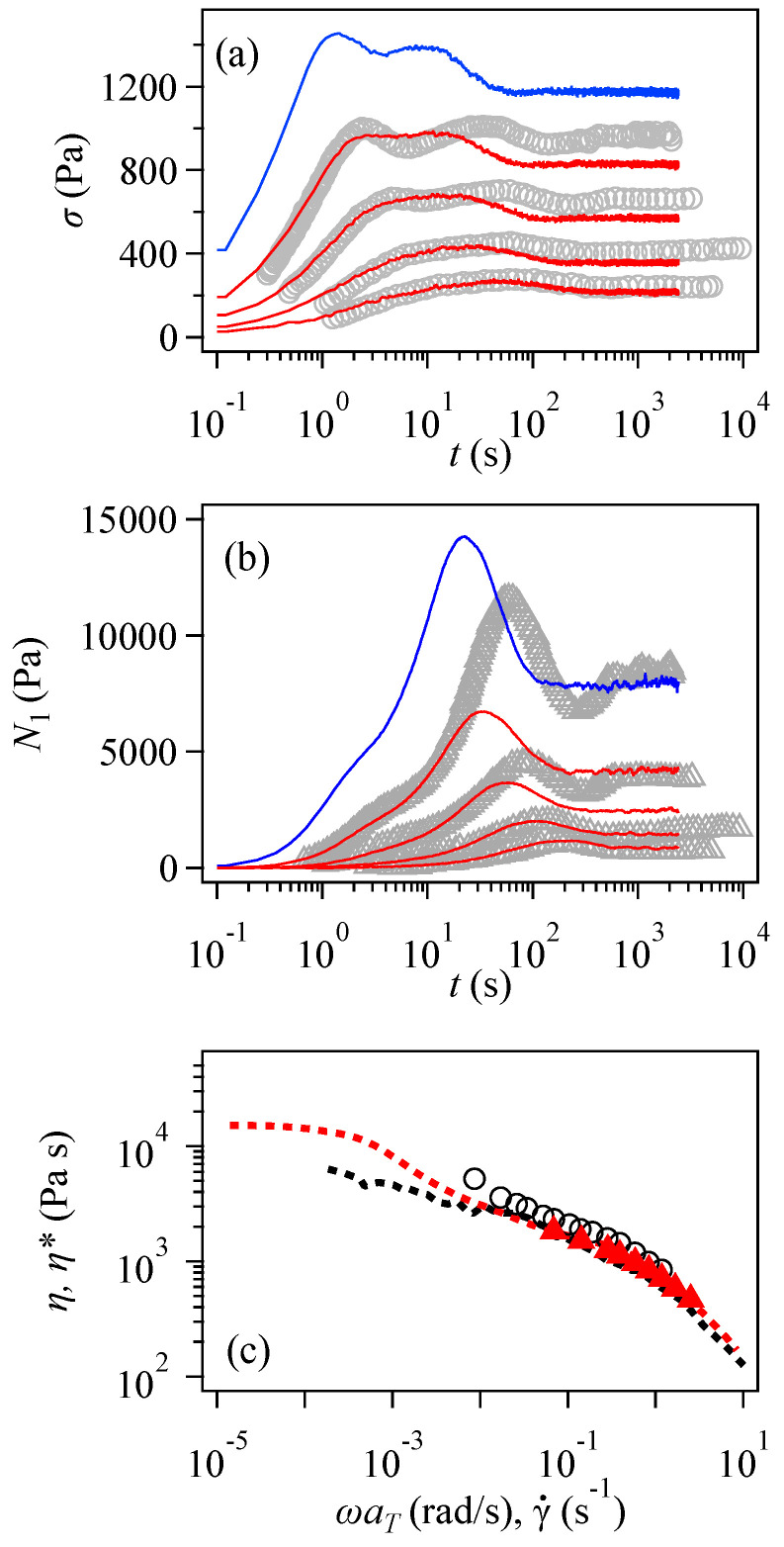
The time development of shear stress (**a**) and the first normal stress difference $N_{1}$ (**b**) at 0 °C for shear rates of 1.165, 0.58, 0.28, and 0.138 s^−1^ from Osaki et al. [15]. These are shown by symbols and compared with our simulation rates, which are indicated by red curves. Blue curves are the simulation results for the shear rate of 2.5 s^−1^, for which experimental data are unavailable. Panel (**c**) shows the steady-state viscosity from Osaki et al. [15] and our simulations, indicated by the black and red symbols, respectively. Black and red broken-line curves indicate the complex viscosity from the experiment and the simulation.

**Figure 4 polymers-16-01455-f004:**
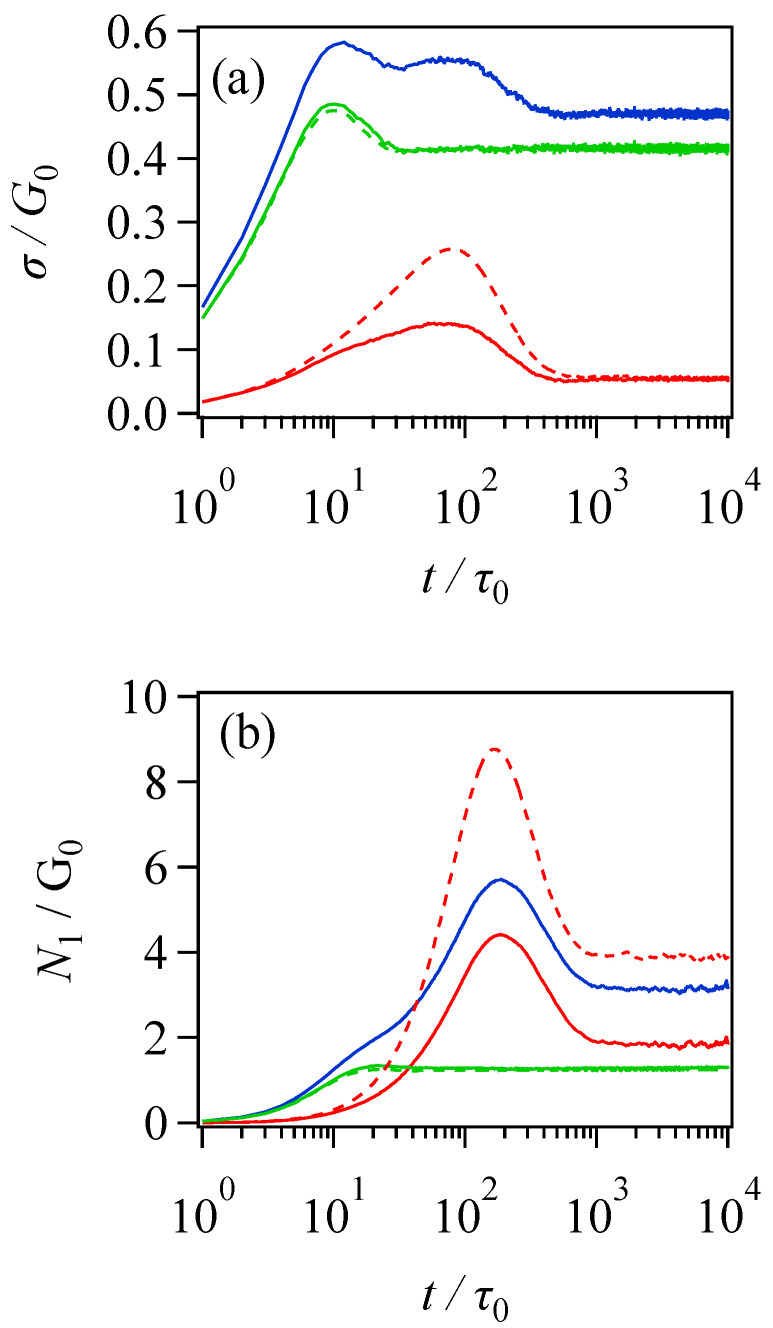
Shear stress (**a**) and the first normal stress difference (**b**) at $\dot{\gamma }\tau_{0}=0.3$. Blue, green, and red curves show the results for the entire system and the contributions from short and long chains. Green and red broken curves indicate the results for pure long- and short-chain systems without mixing, reduced according to their mixing fractions.

**Figure 5 polymers-16-01455-f005:**
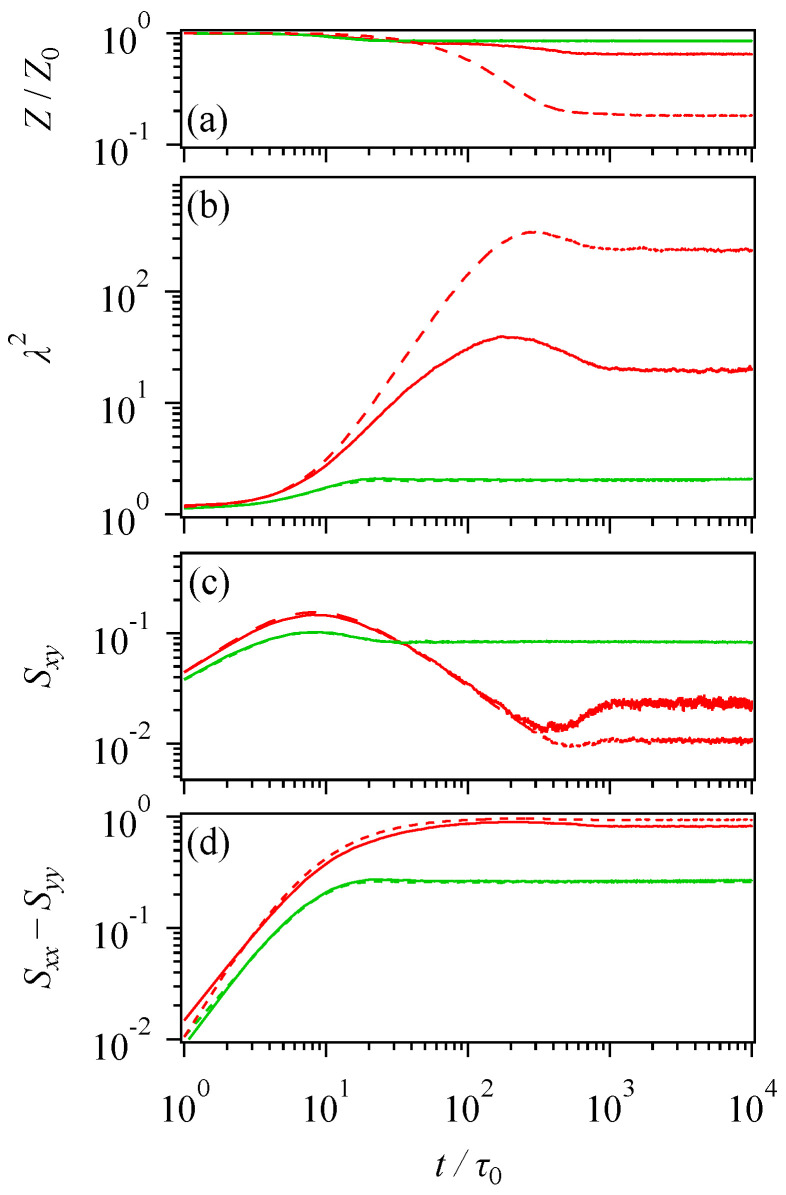
Decoupled analysis of stress concerning the normalized segment number per chain $Z/Z_{0}$ (**a**), the squared segment stretch (**b**), the shear component of orientation tensor (**c**), and the difference between normal orientation components of orientation tensor (**d**). Green and red curves indicate behaviors of the short and long chains, respectively. Solid- and broken- curves display cases with and without mixing of short and long chains. The shear rate applied was $\dot{\gamma }\tau_{0}=0.3$.

## Data Availability

Dataset is not attached as supporting information due to the incompatible format and it will be available upon request from the author.

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
