# Peer review of "Primitive Chain Network Simulations for Double Peaks in Shear Stress under Fast Flows of Bidisperse Entangled Polymers"

_polymers, 2024, doi:10.3390/polym16111455_

Round 1

Reviewer 1 Report

Comments and Suggestions for Authors

In the manuscript, the author presents shear stress results from a primitive chain representation of polydisperse entangled linear polymers and contrasts these simulation results with experimental data reported on earlier in the literature. With the aim to understand a doubly peaked stress response that has been shown to occur during high-rate shear in polystyrene, the author uses simulation to decompose the stress response into contributions originating from short-chain orientation and long-chain stretching.  The manuscript is well-written with narrative and analysis of data presented in a concise manner. The questions I have all relate to time and shear rate in the simulations. I am missing or overlooking some key details where further clarification could help me and potentially future readers.

·      As stated in lines 88-89, “deformation and relaxation are repeatedly applied to attain a designated shear rate.”  I am confused by this statement.  Often this simulation approach is referred to as quasi-static and is typically used to somewhat eliminate the time and/or rate dependence of a stress response.  How does this simulation method correlate to a shear rate?  Is the author applying a strain displacement associated with a given shear rate?  If so, what timescale is being used or how is amplitude chosen?

·      I was questioning how molecular simulation results were reported on timescales of seconds as in fig 3. My understanding is that t(s) originates from comparisons between simulation and experimental G’ and G’’ as shown in Figure 2. Is this correct? Further explanation would be valuable. What is the amplitude of the experimental oscillation? Line 108 was confusing to me. Is tau_0 really a fit to the data? Given simulations are performed in dimensionless units, one could anticipate a scaling of tau_0 such that sim and exp agree even better in fig 2.

·      What implications do the discrepancies in fig 2 have on the results presented in fig 3? For example, there was a brief statement about the long timescale relaxations overestimating experimental values (line 134). Does this overestimate mean that the long time, steady state stress values (fig 3) should actually be further out of agreement?  Additionally, what about the underestimate in the loss modulus at high frequencies?

·      Is the double peak behavior only present in shear startup or should one expect this result to occur repetitively during oscillatory shear?  

Minor typo:  Ln 119: “…box size was 16^3, which...”  What units?

Author Response

I thank the reviewer for carefully reading the manuscript and valuable comments. I have seriously considered the inputs to revise the manuscript. Please find my responses to the comments attached. 

Reviewer 2 Report

Comments and Suggestions for Authors

In the manuscript titled “Primitive Chain Network Simulations for Double Peaks in Shear Stress under Fast Flows of Bidisperse Entangled Polymers,” the author studies the molecular origins of double peaks in shear stress observed during fast start-up shear deformations of bidisperse entangled polymers. The phenomenon of double peaks is critical for understanding the non-linear viscoelastic behavior of polymer blends. To investigate this phenomenon, the author uses the Primitive Chain Network model to perform the multi-chain slip-link simulations. The author provides a detailed simulation approach that decomposes the stress contributions of different chain lengths within the polymer blends, providing insights not fully explained by previous studies.

However, the following points should be addressed before publication:

1. The simulation results are compared with the experimental data from Ref. 15, as shown in Figures 2 and 3. However, these experimental data lack a detailed explanation relevant to the system reported in Ref. 15, which doesn’t provide a brief comparative study of the simulation system with two components: 91% ZL = 7 for the short chain and 9% ZH = 82 for the long chain. Moreover, the explanation for the discrepancies between the simulational and experimental results is too weak to be convincing.

2. In Figure 3, the simulation results show that the double peaks obviously appear at the highest shear rate but not at the lower shear rate, which is inconsistent with the experiment data. The author doesn’t provide any explanation for this inconsistency. Does this mean the simulation is insufficient to model the double-peak phenomenon?

3. In line 241, the author discusses the normalized relaxation time for the examined case by using the equation τR/τ0=Z02 /2π2. If the Z0 is 7 and 82 for the short and long chains, respectively, the result should be 2.5 and 341. Please double-check this calculation. If confirmed, the following discussion will need to be corrected accordingly.

4. In lines 144 and 145, please replace 'sigma' with the symbol 'σ' for consistency and clarity in the notation used throughout the paper.

Comments on the Quality of English Language

Some sentences are overly long and complex, potentially confusing readers. Simplifying these could enhance understanding and readability.

Reviewer 3 Report

Comments and Suggestions for Authors

The author describes the primitive chain network (PCN) simulations of the equilibrium and nonlinear shear rheology for a bimodal blend of entangled linear polymers and demonstrates the double stress overshoot peaks at high shear rates. By decomposing the stress into the contribution from two components, the author clearly shows the origin of the two peaks and points out future experimental designs that could verify his hypothesis. The results are also compared with corresponding experimental measurements. Although the simulation results do not agree quantitatively with experimental ones, the simulations successfully capture the important phenomenon (double peaks) and numerical trends in the experiments. This work is scientifically sound and should be of interest to the broad audience of Polymers. Therefore, I recommend publication after addressing the following points:

Major points:

1. Hydrodynamics plays an important role in the dynamics of polymer solutions, and cannot be ignored at time scales shorter than the relaxation time of correlation blobs. Does the author incorporate hydrodynamics in his model? If not, how will it affect the fidelity of simulation results?

2. Finite stretchability is ignored in the simulation due to large M_0. From the lambda^2 data in Fig 5(b), the average segment stretch ratio is not large enough to provoke nonlinear elasticity. However, it is likely that a portion of chains become strongly stretched. It is important to check the segment stretch ratio distribution to ensure the assumption of linear elasticity in the overshoot regime.

Minor points:

1. The author clearly explains the determination of some simulation parameters but misses two important ones: tau_0 and segment density. How are these two parameters determined? 

2. At line 115, is it "mass fraction" or "molar fraction"?

3. The experimental data is not clearly described. Is it f80/f850 from ref 15?

4. At line 138, the author mentions the dependency of solvent quality on temperature. Is it related to time-temperature superposition?

5. The author compares the two-component mixture with pure melts of each component in Figs 3-5. Do the pure melts share the same segment density as the mixture? It is not clearly described in the paper.

6. Figure 2 caption should include a description of the symbol of each dataset.

6. At lines 144-145, change "sigma" to the greek letter.

7. At line 202, please describe how is the segment orientation tensor S calculated.

Round 2

Reviewer 2 Report

Comments and Suggestions for Authors

The author has appropriately addressed all the concerns raised during the previous review. The modifications and clarifications added have significantly improved the clarity and quality of the manuscript.